# The impacts of preoperative frailty on readmission after cardiac implantable electrical device implantation

Tomonori Takeda[1,2]*, Atsuhiro Tsubaki[2,3], Yoshifumi Ikeda[4], Ritsushi Kato[4], Kazuki Hotta[5], Tatsuro Inoue[2,3], Sho Kojima[2,6], Risa Kanai[7], Yoshitaka Terazaki[7], Ryusei Uchida[8], Shigeru Makita[8]

1 Department of Rehabilitation, Saitama Medical University International Medical Center, Hidaka, Saitama, Japan, 2 Graduate School of Niigata University of Health and Welfare, Niigata, Niigata, Japan, 3 Institute for Human Movement and Medical Sciences, Niigata University of Health and Welfare, Niigata, Niigata, Japan, 4 Department of Cardiology, Saitama Medical University International Medical Center, Hidaka, Saitama, Japan, 5 Department of Rehabilitation, Kitasato University School of Allied Health Sciences, Sagamihara, Kanagawa, Japan, 6 Department of Rehabilitation, Kisen Hospital, Katsushika, Tokyo, Japan, 7 Department of Nursing, Saitama Medical University International Medical Center, Hidaka, Saitama, Japan, 8 Department of Cardiac Rehabilitation, Saitama Medical University International Medical Center, Hidaka, Saitama, Japan

* t_takeda@saitama-med.ac.jp, tonsuke.tomo2@gmail.com

**Data Availability Statement:** All relevant data are within the paper.

**Funding:** The author(s) received no specific funding for this work.

## Abstract

Cardiac implantable electrical devices (CIED) such as pacemakers, implantable cardioverter defibrillators, and cardiac resynchronization therapies are generally recommended for older patients and those with severe heart failure (HF). However, there is currently a lack of evidence on the relationship between frailty and readmission rates among patients with CIED. This study investigated whether preoperative frailty influenced readmission rates among patients with CIED over a one-year period following implantation. The study retrospectively analyzed 101 patients who underwent CIED implantations. To compare frailty-based differences in their characteristics and readmission rates, these participants were categorized into frailty and non-frailty groups via the modified frailty index (mFI). The frailty group had a significantly higher readmission rate than the non-frailty group (non-frailty group vs. frailty group = 1 vs. 8 patients: $P < 0.05$). Further, a multivariate analysis showed that frailty was a significant readmission factor. Based on individual analyses with/without histories of HF, the readmission rate also tended to be higher among individuals considered frail via the mFI (readmission rate in HF patients: non-frailty group vs. frailty group = 1 vs. 5 patients: $P = 0.65$; non-HF patients: non-frailty group vs. frailty group = 0 vs. 3 patients: $P = 0.01$). Participants with preoperative frailty showed higher readmission rates within a one-year period following implantation compared to those without preoperative frailty. This tendency was consistent regardless of HF history. The mFI may thus help predict readmission among patients with CIED.

**Competing interests:** Dr. Kato and Dr. Ikeda received grant support from Boston Scientific, Abbot, paid to his institute. Dr. Makita received payment from Daiichi Sankyo Company, Otsuka pharmaceutical, and Fukuda Denshi for lectures, presentations, speakers bureaus, manuscript writing, or educational events for his institute. This does not alter our adherence to PLOS ONE policies on sharing data and materials.

## Introduction

Currently, three highly therapeutic cardiac implantable electrical devices (CIED) are in common usage, including pacemakers (PM) for patients with bradycardia, implantable cardioverter defibrillators (ICD) for the prevention of sudden death, and cardiac resynchronization therapy (CRT) for patients with heart failure (HF) and cardiac dyssynchrony. These treatments are often indicated for older people and/or those with severe HF. In recent years, a new method of left ventricular lead implantation for CRT using mini-thoracotomy or video-assisted thoracoscopy has been reported. In addition, conduction system pacing (e.g., his bandle pacing and left bandle branch pacing) has been presented as an advanced alternative to CRT; however, these methods do not guarantee perfect effect of heart failure therapy [1–3].

In such cases, high fragility is also frequently determined, which may increase the potential for readmission. While different reports have shown similar 30-day readmission rates following CIED implantation in general, at 15.7% [4] and 13% [5], the one-year readmission rate for CRT patients may be as high as 51.7% [6]. However, the factors that contribute to readmission after CIED surgery are currently unclear, and must be better understood to prevent readmission.

Frailty has also been pointed out as an issue that cannot be overlooked in patients with cardiovascular disease [7]. Recently, the average age of CIED patients has increased [8, 9], with a frailty rate of 31% in those with HF [10]. In this context, various studies have shown that frailty affects outcomes in patients with CIED implantation [4, 9, 11, 12], thus contributing to readmission. However, no such studies have investigated preoperative frailty in relation to CIED, meaning there is a lack of evidence on whether preoperative frailty affects postoperative readmission. This issue must be clarified to help increase the quality of life among older patients who are designated for CIED implantation. This study hypothesized that CIED patients with preoperative frailty would have higher postoperative readmission rates than those without frailty.

The modified frailty index (mFI) [13] is commonly used to assess preoperative frailty. Specifically, the mFI is a frailty assessment that is based on the accumulation of physiological defects. A total of 11 items are considered, including (1) non-independence of activities of daily living, (2) history of diabetes mellitus, (3) history of chronic obstructive pulmonary disease, (4) history of congestive heart failure, (5) history of myocardial infarction, (6) history of percutaneous coronary intervention (PCI)/cardiac surgery/angina, (7) hypertension requiring medication, (8) peripheral vascular disease or pain at rest, (9) sensory impairment, (10) transient ischemic attack or cerebrovascular disease without deficits, and (11) cerebrovascular accident with neurological findings. As such, the mFI is centered on the patient's medical history. It has successfully been used to predict post-operative outcomes for patients with bladder cancer [14], gastrointestinal cancer [15, 16], abdominal cancer [17], and vascular disorders [18]. The mFI is a particularly valuable assessment tool due to its ease of use, as it lists only a handful of items with simple indicators.

This study aimed to clarify the ratio of frailty for each device (PM, ICD, and CRT). To test the above hypothesis, this study conducted a retrospective observational analysis to determine whether preoperative frailty, as determined via the mFI was associated readmission within one year following CIED surgery.

## Methods

### Study population and data collection

This was a single-center, case-control study. We enrolled patients with either newly implanted CIED (PM, ICD, CRT) or who underwent battery replacement at our hospital from May 2016

to January 2020. The inclusion criteria were as follows: patients over 65 years of age who underwent de novo CIED implantation or battery replacement and were treated with cardiac rehabilitation. Individuals < 65 years of age and those with missing data were excluded.

For all participants, we collected data on age, sex, diagnosis comorbidities, medical history, echocardiographic findings, laboratory data, operative records, and routine medical procedures from the electronic medical record system. Preoperative frailty was calculated using the mFI, based on medical history at the time of admission and current medical history as recorded in the electronic medical records. We divided the participants into groups according to their mFI results. Based on previous studies, those with two or fewer mFI items were placed into the non-frailty group, while those with three or more items were placed into the frailty group. The primary outcome was defined as follows: readmission for any cause deemed to require inpatient care within one year of treatment; the maximum follow-up period was regulated per the year.

## Statistical analysis

We conducted unpaired t-tests for the continuous variables, which were expressed as means ± standard deviations. We employed Fisher's exact test for the nominal scales, with results expressed as N (%) or medians (min-max). Next, we conducted the Cox proportional hazards regression with one-year readmission set as the objective variable. Based on the outcomes of previous studies, we selected the following dependent variables: sex [19, 20], age at implantation (divided into 75 years of age or older and less than 75 years of age [19, 20]), left ventricular ejection fraction (more than 50%; heart failure with preserved ejection fraction, more than 40%, less than 50%; heart failure with mid-range ejection fraction, and less than 40%; heart failure with reduced ejection fraction [HFrEF] [21], kidney failure (estimated glomerular filtration rate < 60%) [22], anemia (hemoglobin < 10 g/dl) [23], hyponatremia (sodium < 135 mEq/L) [24], ischemic or non-ischemic [25–27], type of CIED [28], type of implant, and frailty [7]. In cases where variables were significant, we created Kaplan-Meier survival curves and conducted intergroup comparisons using Kaplan-Meier estimators with log-rank tests. P-values < 0.05 were considered statistically significant.

We conducted all statistical analyses using EasyR version 1.52 (Saitama Medical Center, Jichi Medical University, Saitama, Japan), which is a graphical user interface for R version 4.02 (The R Foundation for Statistical Computing, Vienna, Austria). More precisely, it is a modified version of the R commander which is designed to add statistical functions that are frequently used in biostatistics [29].

This study was approved by both the Ethics Committees of Saitama Medical University International Medical Center (20–215) and Niigata University of Health and Welfare (18692–210722), and was registered with University hospital Medical Information Network (identification number: 000044925). The requirement of informed consent from patients was waived, and we adopted an opt-out method, which allows patients to express their desire to not participate. It was approved by both ethics committees. The information regarding the use of medical record data for the study and the opt-out method is presented on our hospital's website.

## Results

We initially recruited a total of 121 participants, but 20 were excluded for being younger than 65 years of age. None of them had missing data. As such, the final sample included 101 patients (Fig 1). Table 1 lists the participant characteristics over the study period. As shown, the mean age was 75.8 ± 6.2 years, with 38 (37.4%) female participants. Of all participants, 64 (63.4%) had PM, 20 (19.8%) had ICD, and 17 (16.8%) had CRT. No patients underwent His pacing in

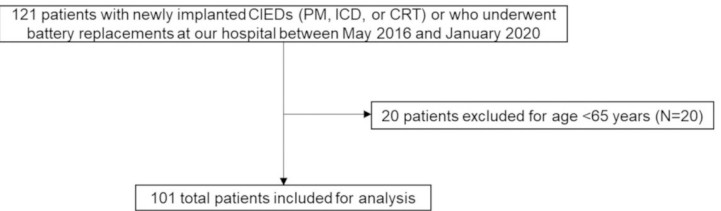

**Fig 1. Flowchart showing the selection of the study population.** CIED = cardiac implantable electrical devices; PM = pacemaker; ICD = implantable cardioverter defibrillator; CRT = cardiac resynchronization therapy.

this study. There were 53 (52.5%) and 48 (47.5%) participants in the non-frailty and frailty groups, respectively. The median of mFI was 2 (0–6).

We found a significant difference in left ventricular ejection fraction between groups (non-frailty group vs. frailty group = 61.2 ± 18.3% vs. 49.3 ± 21.5%, p = 0.004). Looking at frequency, there were also significant intergroup differences in the type of implant, β-blocker, angiotensin-converting enzyme, diuretic, and amiodarone (Table 1). CRT non-responders (defined as improvement in left ventricular end-systolic volume of under 15% from the baseline using echo cardiography at 6 to 12 months after CRT) tended to be more frequent in the frailty group (non-frailty group vs. frailty group = 33.3% vs. 45.5%, p = 1.00). For the subitems of the mFI, there were significant intergroup differences in the non-independence of activities of daily living, diabetes mellitus, history of congestive heart failure, history of myocardial infarction, and history of PCI/cardiac surgery/angina (Fig 2).

A total of nine (8.9%) participants were readmitted within a one-year period, with all causes being the exacerbation of HF; this included one (1.9%) in the non-frailty group and eight (20.8%) in the frailty group. The time to readmission was 68.0 ± 38.7 days overall, with 70.5 ± 40.6 days in the frailty group and 48 days in the non-frailty group. None of the participants died during this period.

Table 2 lists the characteristics of the nine participants who were readmitted within a one-year period. As shown, eight participants from the frailty group (patient numbers 1–8) and one from the non-frailty group (patient number 9) were hospitalized. All were newly implanted with CIED. The average number of days until readmission was 68.0 ± 38.7 days, with the longest being 139 days for patient number 8 for whom all hospitalizations occurred within three months of the operation. Two participants had pacemakers (patient numbers 3 and 4), both of whom were women over 80 years of age with anemia. The average brain natriuretic peptide level was high, while the left ventricular ejection fraction was low; therefore, the population was based on HF.

Table 3 lists the results of the univariate and multivariate analyses conducted to estimate the factors that predicted readmission. As shown, the type of device (high-power device; ICD and CRTD), HFrEF (left ventricular ejection fraction < 40%), and preoperative frailty were significant factors based on the univariate analysis. In addition, frailty was independently associated with the rate of readmission within a one-year period according to the Cox proportional hazards regression analysis. We also generated Kaplan-Meier survival curves, which revealed that frailty was significantly associated with readmission within a one-year period (Fig 3).

We also examined the characteristics of HF and non-HF patients, as we believed the frailty group would have an HF background. Table 4 lists the results of the sub-analysis for HF patients, while Table 5 lists the results of the sub-analysis for non-HF patients. There was a high rate of hospitalization in the frailty group regardless of HF history (readmission rate in HF patients: non-frailty group = 7.1% [n = 1] vs. frailty group = 14.7% [n = 5], p = 0.65; non-HF patients: non-frailty group = 0.0% [n = 0] vs. frailty group = 21.4% [n = 3], p = 0.01).

**Table 1. Participant characteristics.**

| Characteristics | Total (N = 101) | Non-frailty (mFI≤2) (N = 53) | Frailty (mFI≥3) (N = 48) | p value |
|---|---|---|---|---|
| Female | 37 (37.4) | 18 (34.6) | 19 (40.4) | 0.68 |
| Age (years) | 75.9 ± 6.2 | 75.9 ± 6.0 | 75.9 ± 6.5 | 0.96 |
| LVEF (%) | 55.4 ± 20.4 | 61.2 ± 18.3 | 49.3 ± 21.5 | 0.004 |
| **Laboratory data** | | | | |
| BNP (mg/dL) | 318.6 ± 357.9 | 324.4 ± 350.6 | 314.3 ± 370.9 | 0.93 |
| Cr (mg/dL) | 1.5 ± 1.8 | 1.7 ± 2.5 | 1.3 ± 0.6 | 0.43 |
| eGFR (ml/min/1.73m$^2$) | 46.8 ± 20.2 | 50.7 ± 23.6 | 43.3 ± 16.2 | 0.23 |
| HGB (g/dL) | 12.4 ± 1.9 | 12.2 ± 1.7 | 12.6 ± 2.1 | 0.57 |
| Na (mEq/L) | 140.5 ± 2.5 | 141.0 ± 2.0 | 140.0 ± 2.9 | 0.24 |
| **Etiology** | | | | 0.02 |
| SSS | 19 (18.8) | 12 (11.9) | 7 (6.9) | |
| Advanced AVB | 17 (16.8) | 10 (9.9) | 7 (6.9) | |
| CAVB | 32 (31.7) | 20 (19.8) | 12 (25.0) | |
| DCM | 5 (5.0) | 2 (2.0) | 3 (3.0) | |
| HCM | 5 (5.0) | 2 (2.0) | 3 (3.0) | |
| VT/VF | 11 (10.9) | 1 (1.0) | 10 (9.9) | |
| HF | 48 (47.5) | 14 (26.4) | 34 (70.8) | |
| Bradycardic atrial fibrillation | 1 (1.0) | 0 (0.0) | 1 (1.0) | |
| **Type of CIED** | | | | <0.001 |
| PM | 64 (63.4) | 43 (81.1) | 21 (43.8) | |
| ICD | 20 (19.8) | 4 (7.5) | 16 (33.3) | |
| CRT | 17 (16.8) | 6 (11.3) | 11 (22.9) | |
| Non-responder* | 7 (35.3) | 2 (33.3) | 5 (45.5) | 1.00 |
| **Type of treatment** | | | | |
| New CIED implant | 83 (82.2) | 40 (75.5) | 43 (89.6) | 0.07 |
| Battery replacement | 18 (17.8) | 13 (24.5) | 5 (10.4) | |
| **Frailty assessment** | | | | |
| mFI | 2 [0–6] | 1 [0–2] | 3 [3–6] | <0.001 |
| **Time to readmission (days)** | 68.0 ± 38.7 | 48 | 70.5 ± 40.6 | |
| **Treatment in previous surgery** | | | | |
| β-blocker | 42 (41.6) | 14 (13.9) | 28 (27.7) | 0.001 |
| ACE/ARB | 36 (35.6) | 17 (16.8) | 19 (18.8) | 0.53 |
| MRA | 20 (19.8) | 4 (4.0) | 16 (15.8) | 0.002 |
| Diuretic | 39 (38.6) | 10 (9.9) | 29 (28.7) | <0.001 |
| Amiodarone | 18 (17.9) | 3 (3.0) | 15 (14.9) | 0.001 |

The continuous variables are means ± standard deviations. The nominal scales are N (%) or median [min-max].

ACE = angiotensin converting enzyme; advanced AVB = advanced atrioventricular block; ARB = angiotensin II receptor blocker; BNP = brain natriuretic hormone; CAVB = complete atrioventricular block; CIED = cardiac implantable electrical device; Cr = creatinine; CRT = cardiac resynchronization therapy; DCM = dilated cardiomyopathy; eGFR = estimated glemerular filtration rate; HCM = hypertrophic cardiomyopathy; HGB = hemoglobin; HF = heart failure; ICD = implantable cardioverter defibrillator; LVEF = left ventricular ejection fraction; mFI = modified frailty index; MRA = mineralcorticoid receptor antagonist; Na = sodium; SSS = sick sinus syndrome; VT = ventricular tachycardia; VF = ventricular fibrillation.

*CRT non-responder was defined as improvement in left ventricular end-systolic volume of under 15% from the baseline using echo cardiography at 6 to 12 months after CRT.

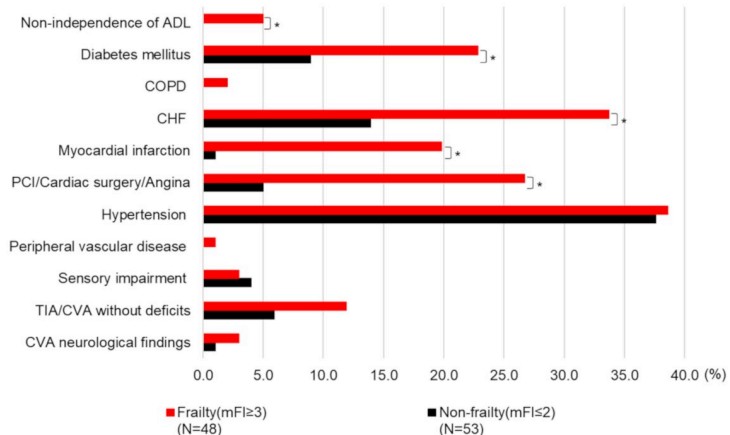

**Fig 2. Comparison of the mFI subitem results between groups.** ADL = activities of daily living; CHF = congestive heart failure; COPD = chronic obstructive pulmonary disease; CVA = cerebral vascular accident; PCI = percutaneous coronary intervention; TIA = transient ischemic attack. Other abbreviations are same as Table 1.

## Discussion

### Major findings

First, the frailty group had a higher readmission rate than the non-frailty group. Indeed, our analysis showed that frailty was a significant predictor for readmission one year after surgery.

Second, the frailty group had significantly higher incidences of several mFI subitems when compared to the non-frailty group, including activities of daily living non-independence, diabetes, HF, myocardial infarction, and PCI/cardiac surgery/angina.

Third, there were many readmissions in the frailty group in each sub-analysis aimed at the presence/absence of HF. In particular, there was a significantly higher rate of readmission in non-HF patients with frailty.

**Table 2. Characteristics of participants who were readmitted.**

| Pt Nr | Sex | Age (years) | mFI | Frailty | Etiology | Type of CIED | Time to readmission (days) | LVEF (%) | BNP (mg/dL) | eGFR (ml/min/1.73m$^2$) | HGB (g/dL) | Type of implant | β-blocker | ACE ARB |
|---|---|---|---|---|---|---|---|---|---|---|---|---|---|---|
| 1 | f | 77 | 3 | yes | VF | ICD | 123 | 26 | 555.6 | 25.2 | 12.2 | New | no | no |
| 2 | m | 75 | 3 | yes | HF/ICM | ICD | 83 | 43 | 163.4 | 28 | 11.0 | New | yes | yes |
| 3 | f | 83 | 4 | yes | Advanced AVB | PM | 56 | 70 | 52.6 | 40.8 | 9.4 | New | no | no |
| 4 | f | 88 | 3 | yes | Bradycardiac AF | PM | 33 | 52 | 166.3 | 36.5 | 9.6 | New | no | no |
| 5 | m | 68 | 3 | yes | VT/VF | ICD | 36 | 20 | 525 | 75.8 | 12.5 | New | yes | no |
| 6 | f | 75 | 5 | yes | CAVB | CRT-D | 46 | 34 | 937.4 | 31.7 | 12.0 | New | yes | yes |
| 7 | m | 73 | 3 | yes | HF/DCM | ICD | 48 | 19 | 304.8 | 61.8 | 10.7 | New | yes | no |
| 8 | m | 66 | 3 | yes | CHF | CRT-D | 139 | 20 | 369.4 | 66.2 | 13.9 | New | no | no |
| 9 | m | 78 | 2 | no | DCM/NSVT | CRT-D | 48 | 11 | 2334.9 | 23.9 | 12.9 | New | yes | yes |
| Av ±Sd | | 75.9±6.8 | | | | | 68.0±38.7 | 32.7 ±19.0 | 601.0 ±702.7 | 43.3±19.5 | 11.6 ±1.5 | | | |

AF = atrial fibrillation: Av = average; CRT-D = cardiac resynchronization therapy with defibrillator; f = female; ICM = ischemic cardio myopathy: m = male:

NSVT = non sustained ventricular tachycardia

Other abbreviations are same as Table 1.

**Table 3. Results of the Cox proportional hazards regression.**

| | Univariate analysis | | | | Multivariate analysis | | | |
|---|---|---|---|---|---|---|---|---|
| | **Wald test** | **HR** | **95%CI** | **p value** | **Wald test** | **HR** | **95%CI** | **p value** |
| Female | 0.19 | 0.74 | 0.19–2.76 | 0.65 | 12.22 | | | |
| Age ≥ 75 years | 0.23 | 1.39 | 0.34–5.57 | 0.63 | | | | |
| HFmrEF | 0.04 | 1.24 | 0.15–9.97 | 0.83 | | | | |
| HFrEF | 0.04 | 5.55 | 1.38–22.21 | 0.01 | | 3.96 | 0.97–16.11 | 0.05 |
| Kidney failure | 0.01 | 1.08 | 0.27–4.34 | 0.90 | | | | |
| Anemia | 2.23 | 3.31 | 0.68–15.91 | 0.13 | | | | |
| Hyponatremia | 0.00 | 0.00 | 0-Inf | 0.99 | | | | |
| Ischemic | 2.00 | 2.58 | 0.69–9.62 | 0.15 | | | | |
| Type of CIED | 5.30 | 6.33 | 1.20–35.88 | 0.02 | | 1.79 | 0.17–18.31 | 0.62 |
| Battery replacement | 0.00 | 0.00 | 0-Inf | 0.99 | | | | |
| Frailty | 4.42 | 9.30 | 1.16–74.42 | 0.03 | | 6.92 | 0.84–56.62 | 0.03 |

CI = confidence interval; HFmrEF = heart failure with mid-range ejection fraction; HFrEF = heart failure with reduced ejection fraction; HR = hazard ratio.

Other abbreviations are same as Table 1.

## Relationship between frailty and readmission in CIED patients

We investigated whether preoperative frailty affected postoperative readmission within a one-year period following implantation in patients with CIED. Our initial hypothesis was that CIED patients with preoperative frailty would have a higher postoperative readmission rate than those without frailty. In the study sample, 20% of CIED patients with preoperative frailty

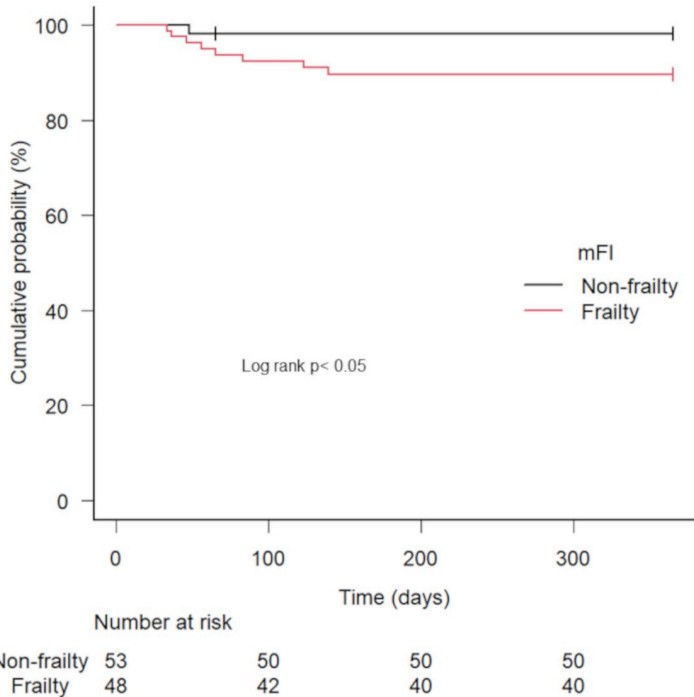

**Fig 3. Kaplan-Meier curve comparing time to development for uncompensated heart failure in patients with cardiac implantable electrical devices (CIED), as classified according to frailty condition, adjusted HFrEF, and CIED type.**

**Table 4. Characteristics of participants with heart failure.**

| | Total (n = 48) | Non-frailty (n = 14) | Frailty (n = 34) | p value |
|---|---|---|---|---|
| Female (%) | 20 (41.7) | 5 (35.7) | 15 (44.1) | 0.75 |
| Age (years) | 76.0 ± 6.5 | 76.36 ± 6.3 | 75.9 ± 6.7 | 0.81 |
| LVEF (%) | 47.2 ± 21.2 | 40.1 ± 19.3 | 49.3 ± 21.5 | 0.19 |
| **Laboratory data** | | | | |
| BNP (mg/dL) | 469.8 ± 531.3 | 570.7 ± 638.3 | 427.0 ± 483.7 | 0.40 |
| Cr (mg/dL) | 1.4 ± 1.2 | 1.8 ±1.9 | 1.2 ± 0.6 | 0.12 |
| eGFR (ml/min/1.73m$^2$) | 45.6 ± 19.8 | 44.5 ± 25.3 | 46.1 ± 17.4 | 0.81 |
| HGB (g/dL) | 12.3 ± 1.8 | 12.0 ± 1.9 | 12.4 ± 1.8 | 0.46 |
| Na (mEq/L) | 139.8 ± 2.7 | 140.1 ± 1.9 | 139.7 ± 2.9 | 0.62 |
| **Etiology** | | | | 0.37 |
| SSS | 6 (12.5) | 1 (7.1) | 5 (14.7) | |
| Advanced AVB | 5 (10.4) | 1 (7.1) | 4 (11.8) | |
| CAVB | 9 (18.8) | 2 (14.3) | 7 (20.6) | |
| DCM | 5 (10.4) | 3 (21.4) | 2 (5.9) | |
| HCM | 5 (10.4) | 2 (14.3) | 3 (8.8) | |
| VT/VF | 6 (12.5) | 0 (0.0) | 6 (17.6) | |
| HF | 48 (100) | 14 (100) | 34 (100) | |
| Bradycardic atrial fibrillation | 1 (2.1) | 0 (0.0) | 1 (2.9) | |
| **Type of device (%)** | | | | |
| PM | 17 (35.4) | 4 (28.6) | 13 (38.2) | 0.74 |
| CRT, ICD | 31 (64.6) | 10 (71.4) | 21 (61.8) | |
| **Type of implant (%)** | | | | |
| New CIED implant | 39 (81.2) | 10 (71.4) | 29 (85.3) | 0.42 |
| Battery replacement | 9 (18.8) | 4 (28.6) | 5 (14.7) | |
| **Frailty assessment** | | | | |
| mFI | 3 [2–4] | 2 [2–2] | 3.50 [3–4] | <0.001 |
| **Treatment in previous surgery** | | | | |
| β-blocker (%) | 32 (66.7) | 9 (64.3) | 23 (67.6) | 1.00 |
| ACE/ARB | 25 (52.1) | 8 (57.1) | 17 (50.0) | 0.75 |
| MRA | 13 (27.1) | 3 (21.4) | 10 (29.4) | 0.72 |
| Diuretic (%) | 26 (54.2) | 6 (42.9) | 20 (58.8) | 0.36 |
| Amiodarone (%) | 13 (27.1) | 2 (14.3) | 11 (32.4) | 0.29 |
| Readmission (%) | 6 (29.1) | 1 (7.1) | 5 (14.7) | 0.65 |
| Time to readmission (days) | 61.2 ± 39.7 | 48 | 63.8 ± 43.8 | 0.65 |

The continuous variables are means ± standard deviations. The nominal scales are N (%) or median [min-max].

Abbreviations are same as Table 1.

were readmitted for worsening HF within a one-year period following surgery. We also conducted univariate and multivariate regression analyses, the first of which included sex, age, cardiac function, renal function, anemia, hyponatremia, ischemia, type of CIED, and new implantation, and the second of which included cardiac function, type of CIED, and frailty as covariates. Patients' sex has important research implications for frailty. For example, female frailty patients with HF have been shown to be at a higher risk of readmission [30]. However, in this study, the percentage of women was low as the subject of this study was heart disease. In addition, since mFI is an index that does not include sex, we consider sex to have had little effect on this study. The results of our multivariate analysis showed that frailty was significantly

**Table 5. Characteristics of participants without heart failure.**

| | Total (n = 53) | Non-frailty (n = 39) | Frailty (n = 14) | p value |
|---|---|---|---|---|
| Female (%) | 18 (34.0) | 13 (33.3) | 5 (35.7) | 1.00 |
| Age (years) | 75.72 ± 5.8 | 75.7 ± 5.84) | 75.7 ± 5.8 | 1.00 |
| LVEF (%) | 64.5 ± 16.3 | 67.2 ± 13.3 | 56.4 ± 21.5 | 0.03 |
| **Laboratory data** | | | | |
| BNP (mg/dL) | 238.4 ± 362.3 | 201.8 ± 314.0 | 330.0 ± 462.6 | 0.27 |
| Cr (mg/dL) | 1.5 ± 1.9 | 1.2 ± 1.9 | 1.9 ± 1.9 | 0.31 |
| eGFR (ml/min/1.73m$^2$) | 53.3 ± 22.1 | 58.65 ± 20.6 | 38.3 ± 19.7 | 0.00 |
| HGB (g/dL) | 12.9 ± 1.8 | 13.3 ± 1.7 | 11.9 ± 1.9 | 0.02 |
| Na (mEq/L) | 140.3 ± 3.0 | 140.5 ± 2.8 | 139.8 ± 3.6 | 0.46 |
| **Etiology** | | | | 0.02 |
| SSS | 13 (24.5) | 11 (28.2) | 2 (14.3) | |
| Advanced AVB | 12 (22.6) | 9 (23.1) | 3 (21.4) | |
| CAVB | 22 (41.5) | 18 (46.2) | 5 (35.7) | |
| VT/VF | 5 (9.4) | 1 (2.6) | 4 (28.6) | |
| **Type of device (%)** | | | | |
| PM | 47 (88.7) | 38 (97.4) | 9 (64.3) | 0.004 |
| CRT, ICD | 6 (11.3) | 1 (2.6) | 5 (35.7) | |
| **Type of implant (%)** | | | | 0.42 |
| New CIED implant | 44 (83.0) | 31 (79.5) | 13 (92.9) | |
| Battery replacement | 9 (17.0) | 8 (20.5) | 1 (7.1) | |
| **Frailty assessment** | | | | |
| mFI | 2 [1–3] | 1 [1–2] | 3 [3–3] | <0.001 |
| **Treatment previous surgery** | | | | |
| β-blocker (%) | 10 (18.9) | 5 (12.8) | 5 (35.7) | 0.10 |
| ACE/ARB | 11 (20.8) | 9 (23.1) | 2 (14.3) | 0.70 |
| MRA | 7 (13.2) | 1 (2.6) | 6 (42.9) | 0.001 |
| Diuretic (%) | 13 (24.5) | 4 (10.3) | 9 (64.3) | <0.001 |
| Amiodarone (%) | 5 (9.4) | 1 (2.6) | 4 (28.6) | 0.01 |
| Readmission (%) | 3 (5.6) | 0 (0.0) | 3 (21.4) | 0.01 |
| Time to readmission (days) | 87.3 ± 33.7 | no patient | 87.3 ± 33.7 | - |

The continuous variables are means ± standard deviations. The nominal scales are N (%) or median [min-max].

Abbreviations are same as Table 1.

associated with readmission within a one-year period. In sum, these findings provide valuable insights for early efforts aimed at preventing readmission, which may be predicted prior to CIED surgery based on mFI results.

In comparison to the 20% of participants with frailty who were readmitted for worsening HF within one-year in this study, a previous study reported that 51.7% of participants who underwent CRT surgery were readmitted for HF within the same timeframe [6]. We believe this difference is due to variations in the participant characteristics and method of determining frailty. For example, the previous study was limited to patients with CRT, while this study considered a wider range of CIED, including PM, ICD, and CRT. In general, patients with PM and ICD have a lower rate of HF readmission and mortality when compared to CRT patients. The potential impact of physical and cognitive dysfunction on CRT is higher than those of PM and ICD. Frailty has been indicated as being an effect on readmission in CRT patients [13]. In

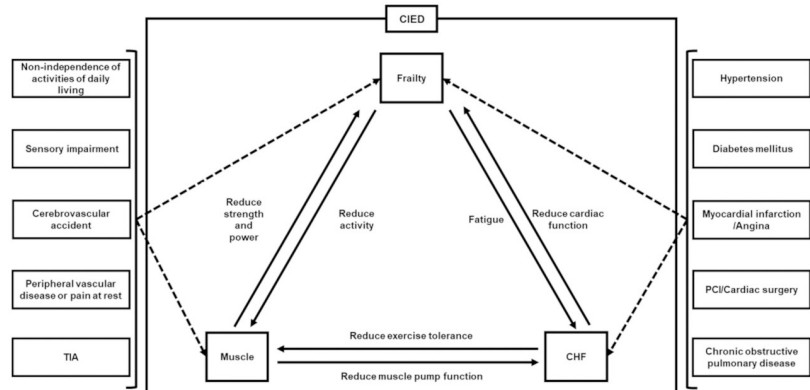

**Fig 4. Correlation chart for frailty, heart failure, CIED patients, and other mFI factors.** Frailty and muscle are affected by ADL, sensory impairment, CVD, PVD, and TIA. Frailty and CHF are affected by hypertension, diabetes mellitus, MI, anginal pain, PCI, and cardiac surgery. Muscle and frailty are associated with decreased muscle mass and strength and decreased activity. Muscle and CHF are related to exercise tolerance and muscle pump action. Frailty and CHF are related to cardiac dysfunction and fatigue. Abbreviations are the same as in Table 1 and Fig 1.

particular, it has been reported that frailty was more frequent in non-responders to CRT [31]. Furthermore, there is literature suggesting a cognitive impairment in CRT patients [32]. Cognitive dysfunction has been an independent factor that increases readmission in heart failure [33]. In this study, only 16.8% of the current participants had CRT. When we added an investigation of the CRT non-responder rate in frailty and non-frailty group, we found that the rate of non-responders tended to be high in the frailty group, but there was no significant difference. Non-responders of CRT may be a cause of readmission for heart failure. CRT non-responders and frailty are covariates and difficult to separate because of a small number of objects. In addition, cognitive function was not investigated in detail in this study, but it may also attenuate the effects of CRT in the frailty population. However, we also found that frailty influenced one-year readmission rates after CIED surgery even after including cardiac function and the type of CIED as covariates.

Frailty, muscle, and CHF have a dynamic interrelation (Fig 4). Frailty and muscle are affected by activities of daily life, sensory impairment, cerebrovascular accident, peripheral vascular disease, and TIA. Some CIED patients with a background of cardiac disease are more prone to frailty. Hypertension, diabetes mellitus, myocardial infarction, angina, PCI/cardiac surgery, and chronic obstructive pulmonary disease cause frailty and are also direct causes of heart failure. Moreover, these induce organic disorder, which causes heart failure. Patients with frailty prior to CIED implantation have reduced activity, muscle strength, and power, which could further exacerbate heart failure after implantation. Reduced muscle mass or muscle function decreases muscle pump function and increases the risk of heart failure. Heart failure not only reduces cardiac function, but also contributes to reduced exercise tolerance, making the patient more susceptible to fatigue, which further contributes to heart failure symptoms.

## Usefulness and bias of mFI

While this study used the mFI to determine frailty, previous studies have used the Fried method [34]. Of note, the mFI can be used to determine the presence or absence of frailty based on medical records; that is, without the need for special equipment. Since the Fried frailty assessment [34] requires a walking speed measurement, it is not considered suitable for

preoperative screening among CIED patients who are at risk of syncope or worsening HF. On the other hand, the mFI is both practical and safe for predicting one-year readmission among this population. However, it is worth mentioning that the mFI subitems are biased toward information related to heart disease. Further, very few studies have investigated the effects of frailty on readmission rates within one year after CIED surgery, which highlights the need for continued investigation.

As three of the 11 mFI subitems are related to heart disease, including HF, practitioners and researchers should consider this bias when assessing frailty. For that reason, we examined the characteristics of participants both with and without HF histories. While we believed that frailty would influence readmission in both cases, this tendency was significant in non-HF participants. However, the same participants may have also been affected by renal function, anemia, the type of CIED, and usage of HF medication. Therefore, previous studies may have rather evaluated frailty. Patients with high mFI seemed to be prone to readmission in the hospital with or without a CIED. Future studies should increase the number of patients and consider the differences between patients with and without a CIED. In this study, the multivariate analysis was hindered by the small number of events, which also highlights the need for continued study.

## Clinical implications

This study produced three important results for consideration in the clinical context. First, we found that the mFI could safely be used to predict one-year readmission prior to CIED implantation. Second, we believe that our results could help avoid futile treatment. Third, preoperatively applied mFI can be used to determine the risk of readmission early in the rehabilitation process. This information may be particularly applicable in revealing patients who should be targeted for cardiac rehabilitation, including exercise therapy and lifestyle guidance. Future studies should also search for ways to mitigate fragility in CIED patients, thus reducing the potential for HF readmission.

## Study limitations

This study also has some limitations. For one, its retrospective approach was suitable for predicting one-year readmission rates based on preoperative frailty, but there was no consideration of postoperative status. There was also a limit pertaining to the statistical processing of the results due to the small number of events.

The most noteworthy problem was the differences in patient characteristics between the frailty and non-frailty groups and rates of sex, in addition to the small number of patients and events. The frailty groups had poorer cardiac function, higher rates of ICD and CRT implantation, and higher use of β blockers, mineralocorticoid receptor antagonist, diuretics, and amiodarone; further, this was biased toward the HF population. For that reason, we investigated the effects of frailty on the HF and non-HF groups. In the frailty group, both the HF and non-HF groups showed a tendency toward readmissions, but this was not significant in the HF group. Another difference in patient characteristics is sex. Female frailty patients with HF have been shown to be at a higher risk of readmission [30]. The patients in our study were predominantly male. Thus, we cannot rule out that sex may have influenced readmission. Hence, future studies should predict readmission using the mFI among larger samples, controlling for patient characteristics. Furthermore, it may be necessary to develop an advanced index that excludes the association with heart disease.

In addition, the frailty group had higher rates of diabetes, previous myocardial infarction, and previous PCI/cardiac surgery/previous angina. In this regard, past research has shown

that factors such as diabetes mellitus, myocardial infarction, and history of PCI may increase the risk of HF readmission in HF patients [21], and may thus be confounding factors. However, these studies did not evaluate frailty in this context.

Finally, the mFI is calculated based on items related to the patient's medical history, meaning there is little room for improvement after surgery. This indicates the need for postoperative evaluations aimed at determining whether preoperative or postoperative frailty and cognitive dysfunction impact readmission.

## Conclusions

In this study, preoperative frailty was associated with increased readmission rates over a one-year period following CIED implantation. Moreover, this tendency was consistent regardless of HF history. These findings suggest that the mFI may be highly useful in predicting readmission among patients with CIED.

## Author Contributions

**Conceptualization:** Tomonori Takeda, Atsuhiro Tsubaki, Yoshifumi Ikeda.

**Data curation:** Tomonori Takeda.

**Formal analysis:** Tomonori Takeda, Yoshifumi Ikeda.

**Investigation:** Tomonori Takeda.

**Methodology:** Tomonori Takeda, Atsuhiro Tsubaki, Yoshifumi Ikeda.

**Project administration:** Tomonori Takeda, Yoshifumi Ikeda.

**Resources:** Tomonori Takeda.

**Software:** Tomonori Takeda.

**Supervision:** Atsuhiro Tsubaki.

**Writing – original draft:** Tomonori Takeda.

**Writing – review & editing:** Tomonori Takeda, Yoshifumi Ikeda, Ritsushi Kato, Kazuki Hotta, Tatsuro Inoue, Sho Kojima, Risa Kanai, Yoshitaka Terazaki, Ryusei Uchida, Shigeru Makita.

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
