## [Decision Letter · Decision Letter 0]

14 Jul 2022

PONE-D-22-16133The impacts of preoperative frailty on readmission after cardiac implantable electrical device implantationPLOS ONE

Dear Dr. Takeda,

Thank you for submitting your manuscript to PLOS ONE. After careful consideration, we feel that it has merit but does not fully meet PLOS ONE’s publication criteria as it currently stands. Therefore, we invite you to submit a revised version of the manuscript that addresses the points raised during the review process.

ACADEMIC EDITOR: The manuscript is affected by several shortcomings that do not help to support the conclusions. A paragraph dedicated to limitations of the study should be emphasized. Moreover, the gender impact should be well addressed.

We look forward to receiving your revised manuscript.

Kind regards,

Vincenzo Lionetti, M.D., PhD

Academic Editor

PLOS ONE

Journal Requirements:

Dr. Kato and Dr. Ikeda received grant support from Boston Scientific, Abbot, paid to his institute.

Dr. Makita received payment from Daiichi Sankyo Company, Otsuka pharmaceutical, and Fukuda denshi for lectures, presentations, speakers bureaus, manuscript writing, or educational events for his institute. 

Reviewers' comments:

Reviewer's Responses to Questions

**Comments to the Author**

1. Is the manuscript technically sound, and do the data support the conclusions?

Reviewer #1: Yes

Reviewer #2: Yes

2. Has the statistical analysis been performed appropriately and rigorously? 

Reviewer #1: Yes

Reviewer #2: Yes

3. Have the authors made all data underlying the findings in their manuscript fully available?

Reviewer #1: Yes

Reviewer #2: No

4. Is the manuscript presented in an intelligible fashion and written in standard English?

Reviewer #1: Yes

Reviewer #2: Yes

5. Review Comments to the Author

Reviewer #1: I would congratulate to the patients for this good study, investigating whether preoperative frailty could influence readmission rates among patients with CIED over a one-year period following implantation. Results are extremely interesting, here you find suggestions in order to improve the manuscript:

Introduction: authors state that "Currently, three highly therapeutic cardiac implantable electrical devices (CIED) are in common usage, including pacemakers (PM) for patients with bradycardia, implantable cardioverter defibrillators (ICD) for the prevention of sudden death, and cardiac resynchronization therapy (CRT) for patients with heart failure (HF) and cardiac dyssynchrony" Authors should also discuss alternative techniques for left ventricular pacing in cardiac resynchronization therapy including: surgical access representing the most frequently used as a second choice by either minithoracotomy or especially the video-assisted thoracoscopy (DOI: 10.1111/pace.12320), His pacing (DOI: 10.1111/pace.14336) or Left Bundle Branch Pacing (DOI: 10.3389/fcvm.2021.630399). Please cite this points including 3 fundamental references

Discussion: "In general, patients with PM and ICD have a lower rate of HF readmission and mortality when compared to CRT patients" However, among CRT, authors should also discuss about its potential effects on functional performance and cognition, two determinants of disability, frailty development and survival. Fumagalli et al demostrated that CRT may be associated with higher functional and cognitive profile (DOI: 10.1016/j.ijcard.2016.06.001). These finding could let us hypothesize a powerful effect of treatment to frailty development and may impact the patient population; please discuss in discussion as well as limitation sections

At the same non responders to CRT should be considered. Please discuss and amplify discussion

A nice figure showing mechanisms involved in frailty is definitely welcome

Reviewer #2: In this paper Authors describe how frailty influences the outcome after placing a cardiac stimulator.

The interest of the paper and its relevance are jeopardized by the limited numbers of patients and of events.

Another bias is linked to the mFI that is based on medical history and comorbidities so patients with higher frailty index have higher probability to have readmission independently from the implant of a device.

Otherwise the paper highlight the importance to take in account the frailty in any procedure to avoid futile treatment.

In general the article is well written but of scarse relevance.

6. PLOS authors have the option to publish the peer review history of their article (what does this mean?). If published, this will include your full peer review and any attached files.

Reviewer #1: No

Reviewer #2: No

---

## [Author Response · Author response to Decision Letter 0]

25 Aug 2022

Response to Reviewers’ Comments

We are grateful for being given the opportunity to resubmit our revised manuscript entitled, “The impacts of preoperative frailty on readmission after cardiac implantable electrical device implantation.”

We have incorporated changes that reflect the detailed suggestions made by the reviewers and believe that the revised manuscript has significantly improved as a result. 

For details, please refer to the file "Response to Reviewers".

---

## [Decision Letter · Decision Letter 1]

21 Oct 2022

The impacts of preoperative frailty on readmission after cardiac implantable electrical device implantation

PONE-D-22-16133R1

Dear Dr. Takeda 

We’re pleased to inform you that your manuscript has been judged scientifically suitable for publication and will be formally accepted for publication once it meets all outstanding technical requirements.

Kind regards,

Jaimin R. Trivedi, MBBS, MPH

Academic Editor

PLOS ONE

Additional Editor Comments (optional):

Reviewers' comments:

Reviewer's Responses to Questions

**Comments to the Author**

1. If the authors have adequately addressed your comments raised in a previous round of review and you feel that this manuscript is now acceptable for publication, you may indicate that here to bypass the “Comments to the Author” section, enter your conflict of interest statement in the “Confidential to Editor” section, and submit your "Accept" recommendation.

Reviewer #1: (No Response)

Reviewer #2: All comments have been addressed

Reviewer #3: All comments have been addressed

2. Is the manuscript technically sound, and do the data support the conclusions?

Reviewer #1: Yes

Reviewer #2: No

Reviewer #3: Yes

3. Has the statistical analysis been performed appropriately and rigorously? 

Reviewer #1: Yes

Reviewer #2: Yes

Reviewer #3: Yes

4. Have the authors made all data underlying the findings in their manuscript fully available?

Reviewer #1: Yes

Reviewer #2: Yes

Reviewer #3: Yes

5. Is the manuscript presented in an intelligible fashion and written in standard English?

Reviewer #1: Yes

Reviewer #2: Yes

Reviewer #3: Yes

6. Review Comments to the Author

Reviewer #1: The authors well answered all reviewer's considerations, and finally manuscript changed but definitely improved. Also I would congratulate with all authors for the good results in this very interesting topic

Reviewer #2: Dear Author, unfortunately the number of the events are too low to support any conclusion and any reasonable discussion about the hypothesis

Reviewer #3: It appears that the authors addressed all the reviewers' comments adequately. I have no further questions related to the content of the manuscript.

7. PLOS authors have the option to publish the peer review history of their article (what does this mean?). If published, this will include your full peer review and any attached files.

Reviewer #1: No

Reviewer #2: No

Reviewer #3: No

---

## [Editor Report · Acceptance letter]

27 Oct 2022

PONE-D-22-16133R1 

*The impacts of preoperative frailty on readmission after cardiac implantable electrical device implantation*

Dear Dr. Takeda:

I'm pleased to inform you that your manuscript has been deemed suitable for publication in PLOS ONE. Congratulations! Your manuscript is now with our production department. 

Kind regards, 

on behalf of

Dr. Jaimin R. Trivedi 

Academic Editor

PLOS ONE